# Dental-Plaque Decontamination around Dental Brackets Using Antimicrobial Photodynamic Therapy: An In Vitro Study

**DOI:** 10.3390/ijerph182312847

**Published:** 2021-12-06

**Authors:** Daliana-Emanuela Mocuta (Bojoga), Mariana Ioana Miron, Elena Hogea, Cornelia Muntean, Darinca Carmen Todea

**Affiliations:** 1Department of Oral Rehabilitation and Dental Emergencies, Faculty of Dentistry, “Victor Babes” University of Medicine and Pharmacy, P-ta Eftimie Murgu 2, 300041 Timisoara, Romania; mocuta.daliana@umft.ro (D.-E.M.) todea.darinca@umft.ro (D.C.T.); 2Department of Microbiology, Faculty of Medicine, “Victor Babes” University of Medicine and Pharmacy, Victor Babes Street, No. 16, 300226 Timisoara, Romania; 3Faculty of Industrial Chemistry and Environmental Engineering, Politehnica University Timisoara, Vasile Parvan Street, No. 6, 300223 Timisoara, Romania; cornelia.muntean@upt.ro

**Keywords:** *Streptococcus mutans*, bracket, antimicrobial photodynamic therapy, methylene blue, chlorophyllin–phycocyanin

## Abstract

Background: In orthodontic therapy, the enamel around brackets is very susceptible to bacterial-plaque retention, which represents a risk factor for dental tissues. The aim of this study was to evaluate the effect of methylene blue and a chlorophyllin–phycocyanin mixture, used with and without light activation, in contrast with a 2% chlorhexidine solution, on Streptococcus mutans colonies. Methods: Twenty caries-free human extracted teeth were randomized into five groups. A Streptococcus mutans suspension was inoculated on teeth in groups B, C, D, and E (A was the positive-control group). Bacterial colonies from groups C, D, and E (B was the negative-control group) were subjected to photosensitizers and 2% chlorhexidine solution. For groups C and D, a combined therapy consisting of photosensitizer and light activation was performed. The Streptococcus mutans colonies were counted, and smears were examined with an optical microscope. Two methods of statistical analysis, unidirectional analysis of variance and the Tukey–Kramer test, were used to evaluate the results. Results: A statistically significant reduction in bacterial colonies was detected after the combined therapy was applied for groups C and D, but the most marked bacterial reduction was observed for group D, where a laser-activated chlorophyll–phycocyanin mixture was used. Conclusions: Photodynamic therapy in combination with methylene blue or chlorophyllin–phycocyanin mixture sensitizers induces a statistically significant decrease in the number of bacterial colonies.

## 1. Introduction

*Streptococcus mutans* (*S. mutans*), the most studied pathogenic microorganism of the oral cavity, is known as the main etiological agent of dental demineralization, which leads to dental caries. After a long period of the testing and reevaluation of several methods used to reduce pathogenic oral microorganisms, it can be stated that antimicrobial photodynamic therapy (aPDT) can be adopted as a modality for bacterial decontamination in dentistry [1].

aPDT is a minimally invasive technique based on a combination of a photosensitizer activated by a light source with a specific wavelength to produce singlet oxygen, which inactivates the target oral bacteria [2].

The main applications of photodynamic therapy (PDT) in dentistry have focused on the diagnosis of malignant oral lesions, oral cancer therapy, and the inactivation of target oral bacteria. According to in vitro studies [3], positive results have been reported after the use of aPDT on *S. mutans* biofilms.

In a well-structured dental biofilm, multiple species of oral microorganisms are highly resistant in this complex ecosystem, rarely being observed as a unique species. In the biofilm, microorganisms are less susceptible to antimicrobial substances, which is precisely due to the formation of resistant cell subpopulations. Considering that there are infections produced by the biofilm that occur due to the diversity of microbial species, this bacterial resistance is the result of the horizontal transfer of genes across different types of species [4].

After many studies to assess the resistance of oral microorganisms to antibiotics, antiseptics, and antifungals [5,6,7], it was found that aPDT had not been sufficiently studied, and it may be a viable alternative to classical decontamination therapy.

The main advantages of using PDT according to Nagata et al. [8] are the “supra-protective measures for dental care, oral bacterial decontamination, the low concentration of dyes leading to a low risk of toxicity, the frequency of dyes used in daily practice (methylene blue and erythrosine) for highlighting bacterial plaque, and light sources (halogen lamps and LEDs)”.

Current scientific research focuses on the widespread use of innovative technology. One example is PDT, involving the administration of a photoactive agent (photosensitizer), which, in the presence of visible light and oxygen, releases reactive oxygen, thus leading to the alteration or even death of the target microorganisms [9].

This therapy can be performed with different types of photosensitizers; the most commonly used is methylene blue (MB), whereas a less-studied example is chlorophyllin (CHL). CHL is a semisynthetic porphyrin, a water-soluble food colorant also known as food additive E140, and an antioxidant that protects lipids from peroxidation and proteins from oxidation. Furthermore, it was proven that CHL had antimicrobial effects on Gram-positive pathogens, yeasts, molds, biofilms, and spores [9].

Inconclusive results for antibiotics and antiseptics have been observed in recent decades; for this reason, new classes of photosensitizers (such as phenalenone, porphyrins, chlorides, and curcumin) have been investigated, with favorable results for bacterial decontamination in in vitro studies [10,11].

Many authors have studied the effects of aPDT on pathogenic oral microorganisms, but only a few research projects have focused on fixed orthodontic therapy [4,5,6,7,8]. Orthodontics is a branch of dentistry with continuous challenges in controlling or preventing the demineralization process and white-spot lesions around brackets.

The PDT method has been used for bacterial decontamination for several years; thus, with a well-founded scientific basis, the decontamination of dental plaque in patients with fixed orthodontic appliances can be easy substituted by aPDT [9,10,11].

The design of this in vitro study was based on current trends in dental practice that use specific treatments to reduce the number of oral pathogens. The aim of this study was to evaluate and compare the effects of two sensitizers, methylene blue (MB) and a chlorophyllin–phycocyanin mixture (CHL–PC), on *S. mutans* colonies, with and without light activation, in contrast with a standard approach using a 2% chlorhexidine (CHX) solution. At the beginning of the study, the null hypothesis was that there would be no statistically significant difference between the two types of therapy in the experimental groups.

## 2. Materials and Methods

This study was conducted by the Faculty of Dentistry in collaboration with the Department of Microbiology of the Victor Babes University of Medicine and Pharmacy in Timisoara and the study protocol was evaluated and approved by the Research Ethics Committee (approval number 22 from 28 June 2021).

The protocol comprised several phases: tooth preparation, a calibration procedure for the bacterial suspension and inoculation (performed immediately for all the samples except the positive control group), and aPDT application.

### 2.1. Tooth Preparation

Twenty caries-free monoradicular human teeth extracted for orthodontic reasons, with intact crowns and matured roots, were collected within 2 days and stored in saline solution at 23 °C. After selecting the teeth, ultrasonic scaling (using EMS miniPiezon, SA CH-1260 Nyon Swiss) and toothbrushing (with fluorine-free paste, Clean Polish, Kerr Hawe) were performed, in order to prepare the buccal surfaces of the teeth for the bonding of the metal brackets.

In the same session, the buccal surface of each tooth was demineralized for 30 s using 37% orthophosphoric acid (Ormco Etching Solution, Ormco Corporation, Columbia, SC, USA), which was followed by washing for 20 s and drying, after which the enamel became opaque white, specific to the clinical aspect of enamel demineralization.

Then, the enamel surface was treated with primer (3M, Unitek, Transbond XT, CA, USA), and the metallic brackets (SS Standard 022 Slot, Ortho Classic, USA) were fixed using Transbond Plus Color Change (3M, Unitek, CA, USA) following the manufacturer’s instructions and an LED lamp (420–480 nm, 1500 mV/cm^2^, Rainbow Curing light, China) for 30 s (10 s each for the mesial, distal, and occlusal surfaces).

To ensure the absence of any bacterial form on the tested area, all the probes were sterilized in an autoclave (C306552 ZETACLAVE B 231, Zhemark S.p.A. B class, Italy) at 121 °C for 20 min; then, the samples were collected from around the brackets using a sterile swab.

### 2.2. Bacterial Culturing

The samples were anaerobically incubated on an agar–blood culture medium (Columbia Agar + 5% ram blood, Mediclim, Romania) for 24 h at 37 °C in a thermostat (Jouan IG150 Infrared-Controlled CO_2_ Incubator, Germany). An ATCC 25175 suspension of *S. mutans* obtained from the Microbiology Department of General Medicine was prepared at 0.5 McFarland [12] with a densitometer device (DEN-1 McFarland Densitometer, Biosan, BS-050102-AAF, LATVIA). Eighteen teeth were contaminated with 0.2 mL of *S. mutans* suspension (10^7^ CFU/mL) [13], and all the samples were placed in a sterile container individually for 48 h at 37 °C in a thermostat (Figure 1 and Figure 2).

### 2.3. Work Protocol

The samples were randomized into two control groups (A and B) and three experimental groups (C, D, and E). For the experimental groups (C and D), two consecutive records were obtained. Specifically, the first was obtained after photosensitizer application, and the second was obtained after further photosensitizer application and activation by laser radiation. The groups were as follows:Group A (*n* = 2), positive control (no bacterial inoculation);Group B (*n* = 2), negative control (bacterial inoculation; no treatment);Group C (*n* = 6), 2% methylene blue (Vanelli, Romania) photosensitizer + PDT;Group D (*n* = 6), chlorophyllin–phycocyanin mixture (PhotoActive+, Medical Systems, Germany) + PDT [13];Group E (*n* = 4), 2% chlorhexidine solution (Chlor X 2% Prevest DenPro Limited, India).

After 48 h of incubation, the teeth were subjected to a new sampling phase in the tested area to prove the presence of *S. mutans* colonies. The samples were collected with a sterile swab and incubated anaerobically on agar–blood medium for a 24 h culture at 37 °C in the same thermostat. On the same day, the first plates of culture medium were analyzed using a digital colony counter (Colony Star, MultiLab, Funke Gerber).

The teeth of group C were subjected to 2% MB impregnation for 1 min according to Azizi’s protocol [12]. After that, the teeth were washed with sterile saline solution for 30 s. Samples were collected from the tested area and aerobically stored on agar–blood culture medium for 48 h at 37 °C in a thermostat. Afterward, the colonies of *S. mutans* were counted.

Then, 2% MB was again applied to the tooth surface around the brackets and activated by laser radiation. The irradiation was performed using a diode laser, with a 660 nm wavelength (Kompakt Laser, CL 50-660, Austria), in continuous-wave mode, using a 50 mW output power and an energy density of 3 J/cm^2^ for 60 s per tooth. Samples were collected from the tested area and aerobically stored on agar–blood culture medium for 48 h at 37 °C in a thermostat. Afterward, the colonies of *S. mutans* were counted.

The teeth of group D were exposed to 5% CHL–PC impregnation for 5 min according to the protocol of Lee H.J. et.al. [13] and of Afrasiabi S. et.al. [14]. The CHL–PC mixture was obtained by dissolving the substance in distilled water in order to obtain a solution of 5% concentration. Afterward, the teeth were washed with sterile saline solution for 30 s, and, using a sterile swab, samples from the tested area were collected. All the specimens were aerobically stored on agar–blood culture medium for 48 h at 37 °C in a thermostat; then, the colonies of *S. mutans* were counted.

For the laser activation of group D, the same diode laser was used with the same wavelength, according to the manufacturer’s instructions; it is well known that chlorophyll absorbs red (620–660 nm) and blue light (400–450 nm), and the absorption spectrum of phycocyanin is 580–660 nm (red/yellow) (PhotoActive+, Medical Systems, Germany). The teeth from group D were subjected to 5% CHL–PC impregnation followed by laser activation. The parameters used for the laser radiation were a 660 nm wavelength, continuous-wave mode, a 50 mW output power, and an energy density of 9 J/cm^2^ for 180 s per sample. All the specimens were aerobically stored on agar–blood culture medium for 48 h at 37 °C in a thermostat; then, the colonies of *S. mutans* were counted.

For the teeth of group E, the enamel surface was treated with 2% CHX solution for 30 s, which was followed by rinsing with distilled water and air drying for 5 s [15]. Samples were collected from this group and stored on agar–blood culture medium for 48 h at 37 °C in a thermostat; afterward, the colonies of *S. mutans* were counted.

All the therapeutic interventions were performed on the same day, by the same investigator for all the teeth, under the same environmental conditions, in order to avoid the occurrence of human error and to achieve the standardization of the working protocol. From each experimental tooth, the samples were collected using a sterile swab by brushing the enamel surface before being aerobically incubated for 48 h on agar–blood culture medium at 37 °C in a thermostat.

Then, the streptococcal colony counting was conducted, and the results were reported as CFU/mL (colony-forming units per milliliter) by the same investigator; for accurate observation, smears were examined under an optical microscope (OPTIKA, B-600Tiph, Italy, 100×/1.25× oil PH, PLAN). The statistical analysis of the results was conducted using one-way analysis of variance (ANOVA) and the Tukey–Kramer test for multiple comparisons, and the statistical analysis software used was OriginPro 8 for Microsoft.

## 3. Results

The numerical results were recorded following the evaluation of each examined sample using a digital colony counter (Colony Star, MultiLab, Funke Gerber), as presented in Table 1. This evaluation technique is in line with protocols reported in similar scientific studies, in which the aPDT method was used in association with bacteriological analysis [14,16,17,18].

To report the bacterial counting results, the following numerical values were assigned to all the results expressed in CFU/mL using a logarithmic formula: 10^7^ CFU/mL, 7; 10^6^ CFU/mL, 6; …; 10^1^ CFU/mL, 1. In order to perform the statistical analysis, the groups type variables were recoded with numerical codes as follows: for group A, Level 1 was used; for Group B, Level 2 was used, and so on to group E, for which Level 7 was used (Table 1).

As shown in Table 1, the means and standard deviations were calculated considering the bacterial presence for all the samples from each group after treatment. It can be observed that after the second counting, the mean value for the bacterial colonies in group B (negative control with bacterial inoculation and no treatment) was similar to that of the colonies in group C (MB only). Furthermore, the mean value for the bacterial colonies in group D (CHL–PC only) was higher than that of those in group E. It should be noted that the initial number of bacterial colonies (10^7^ CFU/mL) was the same for all the groups included in the study.

Following the application of the two sensitizers, it was detected that in group C, the mean value for the bacterial colonies decreased to 6.33 from the initial value of 7 (10^7^ CFU/mL), whereas in group D, it decreased to 5.83 from the initial value of 7 (10^7^ CFU/mL). After the combined therapy (sensitizer and laser activation), it was observed that in group C, the mean value for the bacterial colonies decreased to 4.33 from the initial value of 7 (10^7^ CFU/mL), whereas in group D, it decreased to 3.67 from the initial value of 7 (10^7^ CFU/mL). In group E, where only classical therapy with 2% CHX was applied, a mean value of 5.5 for the bacterial colonies was obtained.

Figure 3 shows a graphical representation of the evolution of the number of bacterial colonies following the application of different therapeutic interventions for all the groups considered in the study.

After counting the bacterial colonies, the samples were examined using an optical microscope (OPTIKA, B-600Tiph, Italy, 100×/1.25× oil PH, PLAN) in order to characterize the bacterial colonies. The bacterial colonies were evaluated in two stages: the first after bacterial inoculation for groups B, C, D, and E (Figure 4) and the second after applying all the interventions: MB, MB + PDT, CHL–PC, CHL–PC + PDT, and 2% CHX (Figure 5).

To test the null hypothesis, one-way analysis of variance (ANOVA) was used to compare the means between groups; then, on the basis of the results, the Tukey–Kramer test was applied to establish the differences between groups (Table 2).

The statistical confidence level was set to 95%, and the statistical significance level was set to 5% (*p* < 0.05). The results are shown in Table 2, where for each pair of groups, the Sig parameter was calculated, representing the statistical difference between two groups.

As specified in Table 2, Prob refers to the level of statistical significance (*p*), which was set to 0.05. To compare the different pairs of groups, the Sig parameter was chosen, showing the existence or not of a statistically significant difference between the pairs of compared groups. Sig = 1 indicates a statistically significant difference at a Prob of 0.05, whereas Sig = 0 indicates a statistically insignificant difference at a Prob of 0.05.

Assessing the first pair of groups (negative control group B, i.e., level 2, with positive control group A, i.e., level 1), the value of Sig was 1, showing a statistically significant difference between the mean values for the bacterial colonies for these groups at *p* < 0.05.

A statistically significant difference of *p* < 0.05 was observed while comparing Level 1 vs. 2, 3, 4, 5, 6, 7, and Level 2 vs. 4, 6. The difference between the mean of bacterial colonies for Level 2 vs. 3, 5, 7 and for Level 3 vs. 5, 7 was non-significant.

## 4. Discussion

The aim of this in vitro study was to evaluate the susceptibility of *S. mutans* to two different photosensitizers applied alone and in combination with a light source, in comparison with the classical approach using 2% CHX solution. To this purpose, bacterial colonies counts were performed, and the distribution of colonies at the smears level according to our study design was evaluated.

The results of our study showed that both interventions, the use of MB photosensitizer combined with PDT and the CHL-PC photosensitizer combined with PDT, induced a decrease in the number of bacterial colonies of *S. mutans*. Thus, the null hypothesis can be rejected. The results are in line with other studies from the specialty literature that also showed a reduction in the number of bacterial colonies after using MB and PDT [2,12,16] or CHL-PC and PDT [14].

According to the recorded data, no bacterial microorganisms were detected after sterilization, and hence, the value “0” was assigned to all groups. Consequently, the mean value of bacterial colonies for group A (positive control group) was 0 with a standard deviation of 0, whereas for group B (negative control group), the mean value of bacterial colonies was 7, with a standard deviation of 0.

In group C, the mean value for the bacterial colonies assessed after only applying the MB photosensitizer was 6.33, whereas that after applying the MB photosensitizer combined with PDT was 4.33 (Table 1). According to the statistical analysis (for Level 4 and Level 3, Sig = 1), there was a statistically significant difference between both treatments.

Our results are in agreement with Diniz I.M. et al. [16] who studied the effect of aPDT on dentin discs contaminated with *S. mutans* biofilms. They observed no bacterial colonies reduction for groups treated only with MB or activated only by laser radiation as well as in the control group. The decrease in bacterial colonies was assessed only when they used the combined technique, MB photosensitizer activated by laser radiation.

The results of our study show that the mean values of bacterial colonies recorded in the group treated with MB photosensitizer only were comparable to those obtained from the negative control group. This means that the application of the MB photosensitizer only had the lowest effect on the reduction in number of the bacterial colonies among all the experimental groups. According to the statistical analysis results, the value of the parameter “Sig” is 0 for Level 3 and Level 2 in pairs, which means that there is no significant statistical difference at *p* = 0.05.

In group D, the mean value for the bacterial colonies assessed after only applying the CHL–PC photosensitizer was 5.83, whereas that after applying the CHL–PC photosensitizer combined with PDT was 3.67 (Table 1). According to the statistical analysis (Sig = 1), there was a statistically significant difference between the two treatments. The lowest mean value for the bacterial colonies was recorded for group D after applying the combined therapy (CHL–PC photosensitizer + laser radiation).

Our findings regarding the reduction of the number of bacterial colonies, after using the combined therapy (photosensitizers and laser radiation), are consistent with other studies in the specialty literature [15,16,17,18,19,20]. Although there are many “in vitro” studies focused on this topic, only a few clinical applications of aPDT are known for oral bacterial decontamination and treatment of the decay disease [19,20,21,22]. 

In their study, Daliri F. et al. [23] evaluated the efficiency of the laser, methylene blue, and curcumin photosensitizer and their associations’ effects on bacterial colonies. They obtained better results on the group where the laser and curcumin intervention was used than on the group where laser and MB was applied. However, the limitation of their study was that they used *candida albicans* as bacterial colonies samples.

Saeed H.M.M. et al. [18] conducted an “in vivo” study where they evaluated the effect of MB activated by laser radiation on superficial and deep dentin using the aPDT therapy. The results of their study showed a reduction in bacterial colonies number but without significant statistical difference between the irradiated dentin levels.

In group E, which received only 2% CHX solution (classical approach), the mean value for the bacterial colonies was 5.5, which is similar to that for group D after only applying the CHL–PC photosensitizer. Accordingly, no statistically significant difference (Sig = 0) was recorded between group E (5.5) and groups C and D after the first determination (C: 6.33 (only MB); D: 5.83 (only CHL-PC)) (Table 2).

Therefore, the null hypothesis suggesting no significant difference between the two types of therapy in the experimental groups can be rejected.

In our study, we did not obtain a statistically significant difference (at *p* = 0.05) in the reduction of the number of bacterial colonies between the study groups and group 5 (where only CHX was applied). These results are in contradiction to those obtained by Arash Azizi et al. [24], who achieved a statistically significant bacterial decrease when they used combined therapy—PDT associated with CUR or PDT associated with MB—in comparison to the control group (CHX).

Currently, in many interventions performed at the level of the oral cavity (periodontal, surgery, orthodontic, and endodontic), therapies based on CHX solution are the gold standard for oral bacterial decontamination, representing an affordable treatment all around the world [25].

At present, most patients use 0.2% CHX solution as an antiplaque agent. Our results are in accordance with Jeyakumar et al. (2020) [26], who stated that this classical approach induced a statistically significant reduction in *S. mutans* colonies in samples collected after 7 days and 14 days of use, but it is not sufficient to state that it is statistically significant compared to the other results.

Leal C.R.L. et al. [27] evaluated the effect of aPDT on the behavior of *S. mutans* from dental plaque, and the results of their study revealed that aPDT in association with MB photosensitizer has the ability to cause DNA damage to bacterial cell. These results trigger us to focus future research in “in vivo” studies on fixed orthodontic therapy in order to obtain the best protocol that could be available both in the dental office and at home.

For in vivo studies focused on oral bacterial decontamination, no validated standard protocol currently exists; therefore, further research on the clinical application of aPDT is needed [18,28].

According to Anselmo et al. [28], combined therapy using PDT and MB led to a statistically significant reduction in bacteria in periodontal pockets, which is consistent with the results of the present study. Moreover, the combined therapy of MB and PDT proves its antibacterial effect both on the pathogenic bacteria responsible for maintaining the periodontal disease and on the bacteria responsible for initiating enamel demineralization (the most important being *S. mutans*). In both studies, the bacterial reduction was assessed using the same radiation wavelength of 660 nm.

The protocol used in our study revealed statistically significant reductions in bacteria between the study groups, as well as between the control and study groups. Accordingly, the null hypothesis was rejected.

However, the design of this study as an “in vitro” experimental evaluation has a few limitations, which may offer directions for future research. In our study, we used groups with low numbers of specimens; therefore, further studies with a larger number of samples are needed. Furthermore, different sets of parameters for laser radiation should be studied. Additionally, the effect of photosensitizers and laser radiation on bacterial plaque from the gingival area should be taken into consideration.

## 5. Conclusions

Within the limitations of this study, the results showed a significant decrease in bacterial colonies when PDT was combined with MB or CHL–PC sensitizers. Considering the advantages of aPDT, the establishment of a valid protocol for decontamination of the dental surface from oral bacterial colonies in orthodontic practice could represent real progress in the quality and efficiency of dental treatments.

## Figures and Tables

**Figure 1 ijerph-18-12847-f001:**
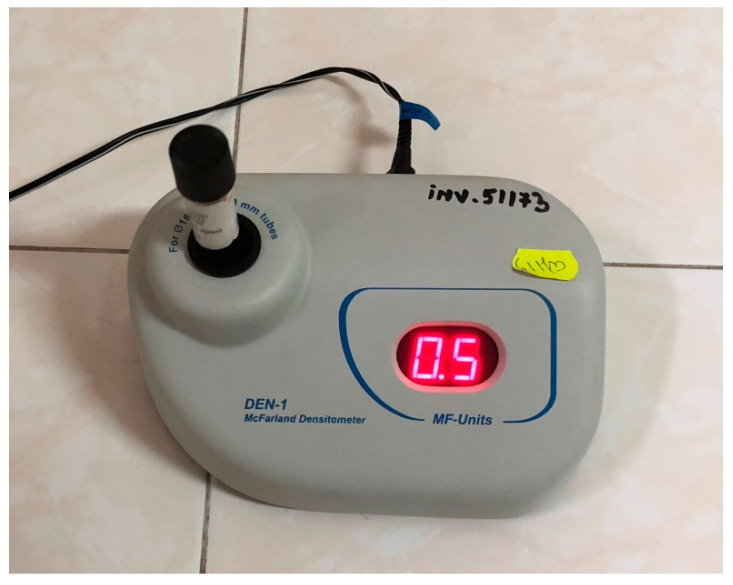
The ATCC 25175 suspension of *S. mutans* prepared at 0.5 McFarland units with the densitometer device (DEN-1 McFarland Densitometer, Biosan, BS-050102-AAF, LATVIA).

**Figure 2 ijerph-18-12847-f002:**
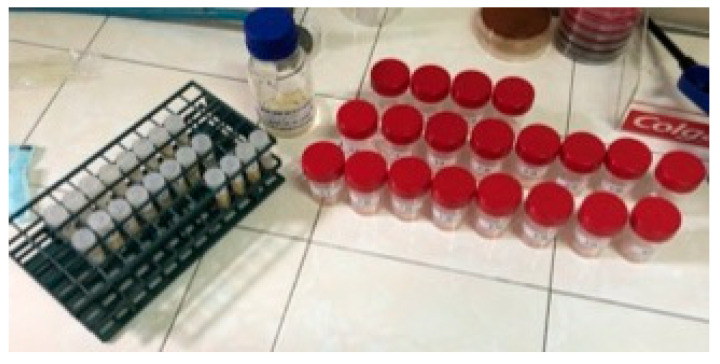
Containers with prepared teeth and metal brackets attached to them, before performing the bacterial inoculation with a 0.2 mL suspension of *Streptococcus mutans*.

**Figure 3 ijerph-18-12847-f003:**
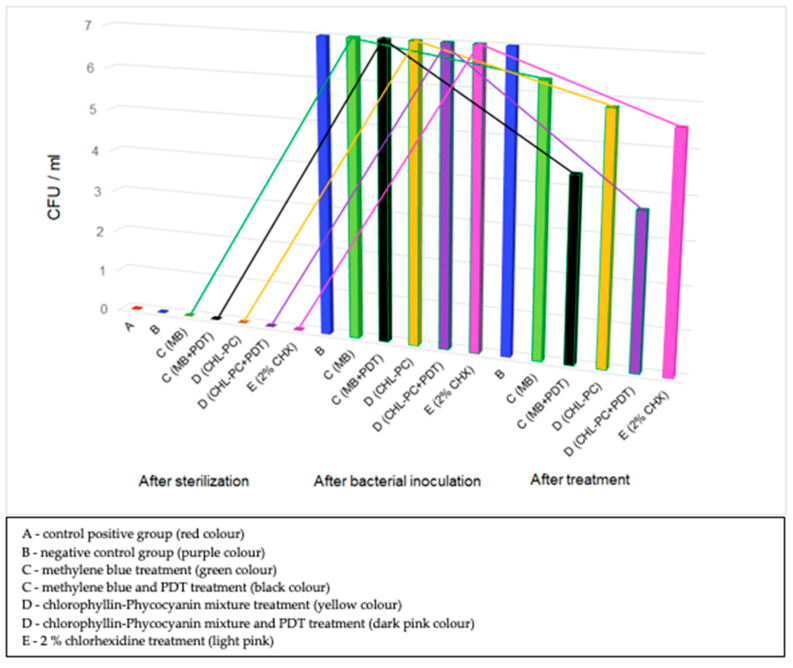
Evolution of the number of bacterial colonies at different stages of the interventional study.

**Figure 4 ijerph-18-12847-f004:**
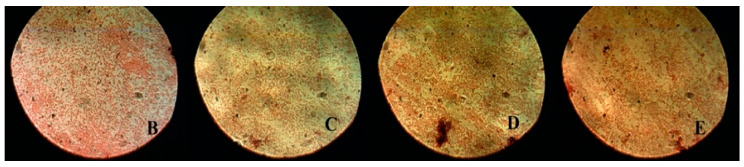
Characterization of bacterial colonies after sample inoculation for groups B, C, D, and E using an optical microscope (OPTIKA, B-600Tiph, Italy, 100×/1.25× oil PH, PLAN).

**Figure 5 ijerph-18-12847-f005:**
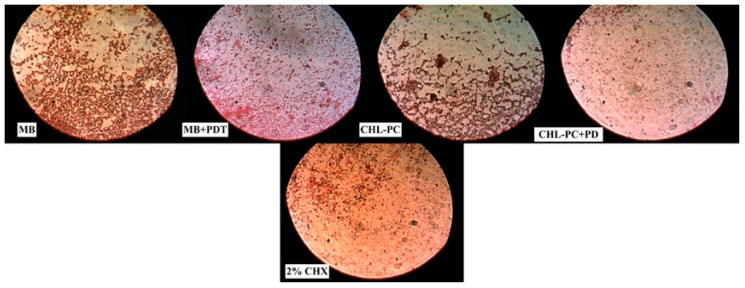
Characterization of bacterial colonies after MB, MB + PDT, CHL–PC, CHL–PC + PDT, and 2% CHX experimental treatments using an optical microscope (OPTIKA, B-600Tiph, Italy, 100×/1.25× oil PH, PLAN).

**Table 1 ijerph-18-12847-t001:** The bacterial colony counts in different phases of the experiment (after sterilizing the samples, after inoculating the bacterial suspension, and after applying the treatments); the minimum and maximum values of the resulting bacterial colonies are reported, along with the means and standard deviations for all groups.

Groups	Equivalent (Level)	Values after Sterilization	Values after Bacterial Inoculation	Values (Minimum)	Values (Maximum)	Mean	SD
**Control+**	Level 1	0	No inoculation	0	0	0	0
**Control−**	Level 2	0	10^7^ CFU/mL	10^7^ CFU/mL	10^7^ CFU/mL	7	0
**MB**	Level 3	0	10^7^ CFU/mL	10^6^ CFU/mL	10^7^ CFU/mL	6.33	0.52
**MB + PDT**	Level 4	-	-	10^3^ CFU/mL	10^6^ CFU/mL	4.33	1.21
**CHL–PC**	Level 5	0	10^7^ CFU/mL	10^5^ CFU/mL	10^7^ CFU/mL	5.83	0.75
**CHL–PC + PDT**	Level 6	-	-	10^2^ CFU/mL	10^5^ CFU/mL	3.67	1.21
**2% CHX**	Level 7	0	10^7^ CFU/mL	10^5^ CFU/mL	10^6^ CFU/mL	5.50	0.58

MB: methylene blue; CHL–PC: chlorophyllin–phycocyanin mixture; CHX: chlorhexidine; CFU/mL: colony-forming units per milliliter; SD: standard deviation.

**Table 2 ijerph-18-12847-t002:** Statistical comparison of pairs of experimental groups taking into account the values of bacterial colonies obtained after treatment, using the Tukey–Kramer test.

Mean Comparisons								
Tukey Test								
	Mean Diff	SEM	*q*-Value	Prob	Alpha	Sig	LCL	UCL
Level 2 Level 1	7	0.89069	11.11438	6.51 x 10^−7^	0.05	1	4.1496	9.8504
Level 3 Level 1	6.33333	0.72725	12.31587	1.36 x 10^−7^	0.05	1	4.00599	8.66068
Level 3 Level 2	−0.66667	0.72725	1.29641	0.96631	0.05	0	−2.99401	1.66068
Level 4 Level 1	4.33333	0.72725	8.42665	5.90 x 10^−5^	0.05	1	2.00599	6.66068
Level 4 Level 2	−2.66667	0.72725	5.18563	0.01734	0.05	1	−4.99401	−0.33932
Level 4 Level 3	−2	0.51424	5.50019	0.01023	0.05	1	−3.64568	−0.35432
Level 5 Level 1	5.83333	0.72725	11.34357	4.61 x 10^−7^	0.05	1	3.50599	8.16068
Level 5 Level 2	−1.16667	0.72725	2.26871	0.68096	0.05	0	−3.49401	1.16068
Level 5 Level 3	−0.5	0.51424	1.37505	0.95547	0.05	0	−2.14568	1.14568
Level 5 Level 4	1.5	0.51424	4.12514	0.09098	0.05	0	−0.14568	1.14568
Level 6 Level 1	3.66667	0.72725	7.13024	5.87 x 10^−4^	0.05	1	1.33932	5.99401
Level 6 Level 2	−3.33333	0.72725	6.48204	0.00186	0.05	1	−5.66068	−1.00599
Level 6 Level 3	−2.66667	0.51424	7.33359	4.09 x 10^−4^	0.05	1	−4.31235	−1.02099
Level 6 Level 4	−0.66667	0.51424	1.8334	0.84717	0.05	0	−2.31235	0.97901
Level 6 Level 5	−2.16667	0.51424	5.95854	0.00465	0.05	1	−3.81235	−0.52099
Level 7 Level 1	5.5	0.77136	10.08368	3.35 x 10^−6^	0.05	1	3.03148	7.96852
Level 7 Level 2	−1.5	0.77136	2.7501	0.47175	0.05	0	−3.96852	0.96852
Level 7 Level 3	−0.83333	0.57494	2.0498	0.77039	0.05	0	−2.67326	1.00659
Level 7 Level 4	1.16667	0.57494	2.86972	0.42234	0.05	0	−0.67326	3.00659
Level 7 Level 5	−0.33333	0.57494	0.81992	0.99688	0.05	0	−2.17326	1.50659
Level 7 Level 6	1.83333	0.57494	4.50956	0.05126	0.05	0	−0.00659	3.67326

Sig = 1 indicates that the mean difference is significant at the 0.05 level. Sig = 0 indicates that the mean difference is not significant at the 0.05 level. The “level” represents the group, and the value is the name of the group, due to software requirements (i.e., level 1 is group A, level 2 is group B, level 3 is group C (methylene blue only), level 4 is group C (methylene blue and laser activation), level 5 is group D (chlorophyllin–phycocyanin mixture only), level 6 is group D (chlorophyllin–phycocyanin mixture and laser activation), and level 7 is group E (2% chlorhexidine)).

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
