# Peer review of "Dental-Plaque Decontamination around Dental Brackets Using Antimicrobial Photodynamic Therapy: An In Vitro Study"

_ijerph, 2021, doi:10.3390/ijerph182312847_

Round 1
Reviewer 1 Report
Congratulations to the authors for all the hard work and time invested in this research. I appreciated the thoroughness with which you explained some difficult concepts. Also, the area of research is of high interest taking into consideration the risks involving orthodontic treatment.
However, there are some aspects that you should keep in mind. These are the following:
- The abstract should not contain the mean values that are presented in the Results section.
- In the Introduction the paragraph from line 45 to line 49 should be rewritten.
- In the Materials and methods section -line 116- all the details regarding the autoclave should be written. Also -line 125- the detailed information regarding the densitometer. For Celsius, for example, the abbreviation °C should be used.
- The terms "Streptococcus mutans" and "in vitro" should be written in italics.
- Regarding the language there some corrections are required. I will write an example- line 38- "which is leading to dental caries". Also, -line 38 "After a long period of testing and re-evaluating several methods that are used to reduce pathogenic oral microorganisms, antimicrobial photodynamic therapy (aPDT) is useful as a modality of bacteria decontamination in the field of dentistry".
- Regarding the figures- for example for figure 1 what are the explanations for each figure. For example a) and b) and each of them could have their own explanation.
- Overall, the presentation could be improved, so that the research and all the steps are presented in a more concise and clear manner.
Again I congratulate all the authors for this interesting and valuable research.
Author Response
We would like to thank you for your appreciation of our work and say we are glad to hear that you found it interesting. We have taken into consideration your recommendations, therefore we have operated the following changes as can be seen in the attached Response:
Kind regards,
The Authors

Reviewer 2 Report
Dear authors,
Congratulations on your work.
I would like to thank you for your efforts in conduncting this interesting study.
However, there are extensive grammar errors in the manuscript and I would make several improvements.
Abstract
- Line 17: It should be “around brackets”
- Line 18: It should be “represents”.
- Line 23 and 24: Please change “Next one, for C and D groups was performed combined therapy:” for “Combined therapy was performed for C and D groups:”.
- Line 26: Please revise this sentence because there are grammar errors.
- Line 27: Please change “are” for “were”.
- Line 28: Please change “are” for “were”.
- Lines 28-31: It should be “Conclusions: Photodynamic Therapy in combination with Methylene Blue or chlorophyllin-phycocyanin mixture sensitizers, induces a statistically significant decrease of the bacterial colonies number.”
Introduction
- Lines 37-38: It should be “Streptococcus mutans (S. mutans), which is the most studied pathogen microorganism of the oral cavity, is known as the main etiological agent of dental demineralisation, which leads to dental caries”.
- Lines 43-44: It should be “in order to produce singlet oxygen for inactivation of the target oral bacteria”.
- Lines 47 and 48: It should be “According to in vitro studies, positive results have been reported after the use of aPDT on S. mutans biofilms”.
- Line 50: Please revise the sentence because there are grammar errors.
- Line 57: Please change “antimicrobial photodynamic therapy” for “aPDT”
- Line 58: Change word “less” with another word, because you do not compare aPDT with another method. I think you mean it has not been studied enough.
- Paragraph 60-64: Please revise the paragraph because there are many grammar mistakes.
- Line 66: It should be “Currently the scientific research focuses on the widespread use of innovative
- One of them is the PDT, which involves……”.
- Line 70: It should be “the most commonly one used being methylene blue (MB) and less studied one being Chlorophyllin (CHL)”.
- Line 73: It should be “protects”.
- Paragraph 76-79: Please revise and reorganize the whole paragraph because there are grammar errors and the meaning is not clear.
- Line 80: It should be: “Many authors have studied the effects of aPDT on pathogenic oral microorganisms, but only few research projects focused on fixed orthodontic therapy”.
- Lines 82-84: It should be “Orthodontics is a branch of dentistry with continuous challenges in controlling or preventing demineralization process and white spot lesions around brackets”.
- Line 80-93: Please add references on the end of each sentence.
- Lines 88-93: Please remove the whole paragraph. You have already explained this in a previouys paragraph (lines 80-84).
- Lines 94-96: Please revise the sentence because there are grammar errors.
- Line 98: Please change “Streptococcus mutans” for “S. mutans”.
- Line 99: It should be “in contrast with a standard approach with 2% Chlorhexidine (CHX) solution”.
- Line 99-101: Please revise the sentence because there are grammar errors.
Material and Methods
- Lines 104-105: It should be “in collaboration with the Microbiology Department at "Victor Babes" University of Medicine and Pharmacy.
- Lines 105-110: Please revise the description of the protocol. There are several grammar errors.
- Lines 112-113: It should be “human extracted teeth for orthodontic reasons, with intact crown and matured roots, were collected and stored…”.
- Please explain in detail what kind of metal brackets and when exactly they were bonded on the teeth.
- Please mention the brand of the ultrasonic scaler and the fluorine free paste.
- Lines 115-116: It should be “the buccal surface for the bonding of metal brackets”.
- Lines 117-119: Please revise this sentence because there are grammar mistakes.
- Line 124: It should be “procured from the Microbiology Department”.
- Lines 125-126: It should be “with a densitometer device (DEN-1 Mc-Farland Densitometer). Eighteen teeth were contaminated with 0,2 mL of S. mutans suspension….”.
- Figure 1: There are 2 images. Please separate them in Figure 1 and Figure 2. Also, please check the description underneath each figure because there are grammar errors.
- Lines 131-133: It should be “For two of the experimental groups (C and D) two consecutive recorders were made. More precisely, the first one was done after a photosensitizer application and the second one was done after a new photosensitizer application and the activation by laser radiation. The groups were the following:”.
- Paragraph 158-179: Please revise the whole paragraph because there are several grammar errors.
- Please add a new paragraph explaining the statistical analysis in detail. There are several grammar errors in your short description of the statistical analysis.
Results
- Line 195: Please remove “same”
- Paragraph 198-201: Please revise the whole paragraph because there are several grammar errors.
- Lines 204-206: Please revise the sentence because there are several grammar errors.
- Lines 209-211: Please revise the sentence because there are several grammar errors.
- Line 228: Please delete “that means the existence of a significant statistically difference between results”.
- Lines 229-231: It should be “The results are shown in Table 2 and for each pair of groups, the "Sig" parameter is calculated, which shows if there is a statistical difference between the two groups.
Discussion
- Line 255: It should be “The results showed that no bacterial life was detected after sterilization”.
- Line 256-258: It should be “Group A (positive control group) was not contaminated with bacterial suspension, while, group B (negative control group) was contaminated with S. mutans suspension, but did not receive any intervention.”.
- Lines 260-261: Please revise this sentence because there are grammar errors.
- Line 263: Change “is” for “was”.
- Lines 264-265: Revise this sentence because there are grammar errors.
- Line 267-268: Change “is” for “was”.
- Lines 268-270: Revise this sentence because there are grammar errors.
- Line 270: Please delete “It can be noticed that”
- Line 273: Change “is” for “was”.
- Lines 273-274: “situated around…other therapies”. Please revise because there are grammar errors.
- Lines 274-277: Revise this sentence because there are grammar errors.
- Lines 278-281: It should be “Therefore, the null hypothesis, that there are no significant differences between the two different types of therapy in the experimental groups, can be rejected.
- Lines 282-286: Please revise the whole paragraph because there are several grammar errors.
- Line 288: Change “are” for “were”.
- Lines 289-291: It should be “This means that the application of only MB photosensitizer had the lowest effect on the bacterial colonies among the experimental groups”.
- Paragraph 292-298: Please remove the last sentence and revise the rest of the paragraph because there are grammar errors.
- Line 306: It should be “The authors”
- Line 307: It should be “a statistically significant”
- Paragraph 310-313: Please revise the paragraph because there are grammar errors.
- Line 317: Please remove “the”
- Paragraph 320-323: Please revise the whole paragraph because there are several grammar errors.
- Paragraph 324-326: Please revise the whole paragraph because there are several grammar errors.
- Paragraphs 327-330: Please revise the whole paragraph because there are several grammar errors.
- Paragraphs 331-336: Please revise the whole paragraph because there are several grammar errors.
- Paragraphs 337-339: Please revise the whole paragraph because there are several grammar errors.
Conclusions
- Line 342: It should be “limitations”.
- Line 342: It should be “significant”.
Author Response
Thank you for rigorously reviewing of our article and your detailed feedback which helps substantially in the improvement of our work. Consequently, we made the following changes which can be seen in the attached Response:
Best wishes,
The Authors

Reviewer 3 Report
Thank you for the opportunity to review this manuscript entitled “Dental plaque decontamination around the dental brackets using the antimicrobial Photodynamic Therapy - an "in vitro" study.” I think that this manuscript was well presented by the authors, and its content may be of interest to the readers. However, the manuscript has several major concerns that must be addressed before publication.
Abstract
・I recommend that the authors revise the method and results because the grouping method and its results were unclear. Moreover, I think that “considering the results of this study” is not necessary.
Materials and Methods
・L111 Tooth preparation
When and how did the authors bond the metallic brackets?
・Please consider correction as you used multiple comparisons.
・I think this control group may be CHX2%. How about you compare B group and C group with E group?
Results
・The authors should move the statistical method to M&M section.
・There was no explanation about table 1.
・L2412-248
 The authors only stated the presence or absence of a significant difference.
Discussion
・The most of your discussion should be mentioned in results section.
Tables and figures
・These are not sufficient. You may add the footnotes and figure legends. In addition, please modify the size to make it easier for the reader to understand.
Author Response
We thank you for revising our article and we are delighted to see that you consider it engaging. In what follows we have taken into consideration your guidelines as can be seen in the attached Response
All the best,
The Authors

Round 2
Reviewer 1 Report
Dear authors, congratulations for your hard work.
I recommend adding a paragraph in the Discussions in order to explain the obtained results. Actually an emphasis on the characteristics of each solution or mixture presented in the Materials and methods in a manner that outlines specifically why the combined therapy PDT and MB determined a statistically reduction of bacteria values from periodontal pockets.
Author Response
Dear Reviewer 1,
We have accepted all the modifications that you suggested. The material was sent to an extensive English revision and on that manuscript we operated the last modifications by taking into consideration all your comments. We thank you again for all your support, implication and professionalism.
With appreciation and gratitude,
All the Authors

Reviewer 2 Report
Thank you for the changes. Congratulations on your work.
Author Response
Dear Reviewer 2,
We have accepted all the modifications that you suggested. The material was sent to an extensive English revision and on that manuscript we operated the last modifications by taking into consideration all your comments. We thank you again for all your support, implication and professionalism.
With appreciation and gratitude,
All the Authors

Reviewer 3 Report
Thank you for submitting revised manuscript. I think the manuscript is improved, however the clarity of presentation, organization, and structure need to be improved. Therefore, I strongly recommend that the manuscript was checked and reconstructed by English native speakers.
Author Response
Dear Reviewer 3,
We have accepted all the modifications that you suggested. The material was sent to an extensive English revision and on that manuscript we operated the last modifications by taking into consideration all your comments. We thank you again for all your support, implication and professionalism.
With appreciation and gratitude,
All the Authors
